# Kinetic prediction of reverse intersystem crossing in organic donor–acceptor molecules

Naoya Aizawa [1,2✉], Yu Harabuchi [2,3,4✉], Satoshi Maeda [3,4] & Yong-Jin Pu [1]

Reverse intersystem crossing (RISC), the uphill spin-flip process from a triplet to a singlet excited state, plays a key role in a wide range of photochemical applications. Understanding and predicting the kinetics of such processes in vastly different molecular structures would facilitate the rational material design. Here, we demonstrate a theoretical expression that successfully reproduces experimental RISC rate constants ranging over five orders of magnitude in twenty different molecules. We show that the spin flip occurs across the singlet–triplet crossing seam involving a higher-lying triplet excited state where the semi-classical Marcus parabola is no longer valid. The present model explains the counterintuitive substitution effects of bromine on the RISC rate constants of previously unknown molecules, providing a predictive tool for material design.

[1] RIKEN Center for Emergent Matter Science (CEMS), 2-1 Hirosawa, Wako, Saitama 351-0198, Japan. [2] Precursory Research for Embryonic Science and Technology (PRESTO), Japan Science and Technology Agency (JST), 4-1-8 Honcho, Kawaguchi, Saitama 332-0012, Japan. [3] Department of Chemistry, Faculty of Science, Hokkaido University, Kita 10, Nishi 8, Kita-ku, Sapporo 060-0810, Japan. [4] Institute for Chemical Reaction Design and Discovery (WPI-ICReDD), Hokkaido University, Kita 21 Nishi 10, Kita-ku, Sapporo, Hokkaido 001-0021, Japan. ✉email: naoya.aizawa@riken.jp; y_harabuchi@sci.hokudai.ac.jp

Electronic spin-flip processes in molecular excited states have attracted increasing interest for optoelectronics[1–3], photo-catalytic synthesis[4–6], and biomedical applications[7–9]. A relevant example is reverse intersystem crossing (RISC), the uphill transition of a non-emissive triplet excited state to an emissive singlet excited state. This process leads to E-type delayed fluorescence, also known as thermally activated delayed fluorescence (TADF), and allows an internal charge-to-photon conversion efficiency of nearly 100% in organic light-emitting diodes[10]. Although materials have typically been discovered experimentally, a fundamental understanding of RISC kinetics and strategy for predicting the rate constants may open vast opportunities for theory-driven materials discovery.

RISC kinetics are often considered in the framework of Marcus theory[11–14]. If the spin–orbit coupling $H_{SO}$ between the initial triplet and final singlet excited states is weak, meaning that the spin-flip only occurs on the crossing seam between their potential energy surfaces (PESs) (Fig. 1a), the RISC rate constant ($k_{RISC}$) follows a Marcus-like nonadiabatic expression:

$$k_{RISC} = \frac{2\pi}{\hbar}\left|H_{SO}\right|^{2}(4\pi\lambda k_{B}T)^{-\frac{1}{2}}\exp\left(\frac{-E_{A}}{k_{B}T}\right) \quad (1)$$

where $\hbar$ is the reduced Planck constant, $k_{B}$ is the Boltzmann constant, $T$ is the temperature, $\lambda$ is the reorganization energy, and $E_{A}$ is the activation energy to reach the crossing seam. In the case of simple parabolic PESs with equal force constants, which is a crucial assumption of Marcus theory, $E_{A}$ can be analytically expressed as

$$E_{A} = \frac{(\Delta E_{ST} + \lambda)^{2}}{4\lambda} \quad (2)$$

with $\Delta E_{ST}$ as the adiabatic singlet–triplet energy difference. A key implication of Eqs. (1) and (2) is that $k_{RISC}$ can be predicted from the equilibrium geometries, which correspond to the easily computable local minima on the PESs of the initial triplet state and final singlet state. However, this understanding of RISC becomes more complicated if the spin-flip process involves an energetically higher-lying excited state as an intermediate[15–19] (Fig. 1b). Since Eq. (2) does not include information on the key intermediate involved in the actual spin-flip process, recent calculations using the equilibrium geometries only provided qualitative justification of the experimental $k_{RISC}$ for a handful of TADF molecules[20]. Herein, we explicitly compute singlet–triplet crossing seams to quantitatively predict $k_{RISC}$ for vastly different structures both from the literature and previously unknown molecules. Rigorous comparisons to experimental data reported over the last decade allowed a general understanding of the RISC kinetics governed by the singlet–triplet crossing seam involving a higher-lying triplet excited state.

## Results

**Computation of $k_{RISC}$.** To understand and predict the RISC kinetics, we first focused on twenty different TADF molecules reported in the literature (Fig. 2a). These molecules are characterized as donor–acceptor systems in which electron-rich donor units, aryl amines, are covalently bound to electron-deficient acceptor units, such as heterocycles, aryl nitriles, ketones, boranes, sulfones, alkynes, or phosphine oxides. We collected literature values for the steady-state and transient photoluminescence data of these molecules and estimated their $k_{RISC}$ values from differential rate equations of the population densities of their singlet and triplet excited states (see Supplementary Information Section 1 for details). The experimental $k_{RISC}$ values varied substantially, by five orders of magnitude, from $10^{2}$ to $10^{7}$ s$^{-1}$.

Directly computing $k_{RISC}$ from Eq. (1) requires the minimum-energy seam of the crossing (MESX), the energetically most accessible geometry on the singlet–triplet crossing seam hypersurface[21,22], as well as equilibrium excited-state geometries. To obtain the MESX for each molecule, we employed a constrained optimization algorithm using the gradient projection method[23], which minimizes the mean energy of the singlet and triplet states $(E_{S} + E_{T})/2$ while simultaneously fulfilling the crossing condition of the energy difference $E_{S} - E_{T} = 0$. $E_{S}$ and $E_{T}$ were calculated at the level of time-dependent density functional theory (TDDFT) within the Tamm–Dancoff approximation[24] (see the Methods for details).

For the MESX geometries of the twenty molecules shown in Fig. 2a, TDDFT predicts nonzero $H_{SO}$ of 0.17–3.61 cm$^{-1}$ with fairly small $E_{A}$ of 0.11–0.32 eV, corresponding to $k_{RISC}$ of $10^{2}$–$10^{7}$ s$^{-1}$ calculated using Eq. (1) at $T$ of 300 K. Figure 2b compares the theoretical $k_{RISC}$ values to the experimental values, demonstrating that the present model successfully reproduces the experimental rates. The mean absolute logarithmic error (MALE) reaches only 0.23, whereas a larger MALE of 1.2, corresponding to an error of 1.2 orders of magnitude, is observed for the values based on the conventional model shown in Fig. 1a and the parabolic approximation of Eq. (2) (see Supplementary Fig. 1 for the errors for each molecule). These results thus demonstrate the importance of the explicit computation of the singlet–triplet crossing seams for quantitatively predicting $k_{RISC}$.

**Mechanism of the RISC.** Closer inspection of the data further reveals that the lowest singlet excited state (S$_1$) does not cross the lowest triplet state (T$_1$) and instead crosses the higher-lying triplet

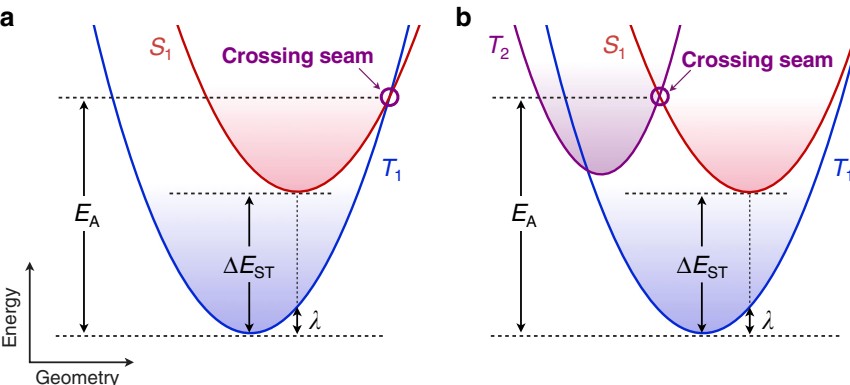

**Fig. 1 RISC from triplet to singlet excited states. a, b** Schematic potential energy surfaces of excited states depicting RISC via (**a**) a S$_1$–T$_1$ crossing seam and (**b**) a S$_1$–T$_2$ crossing seam.

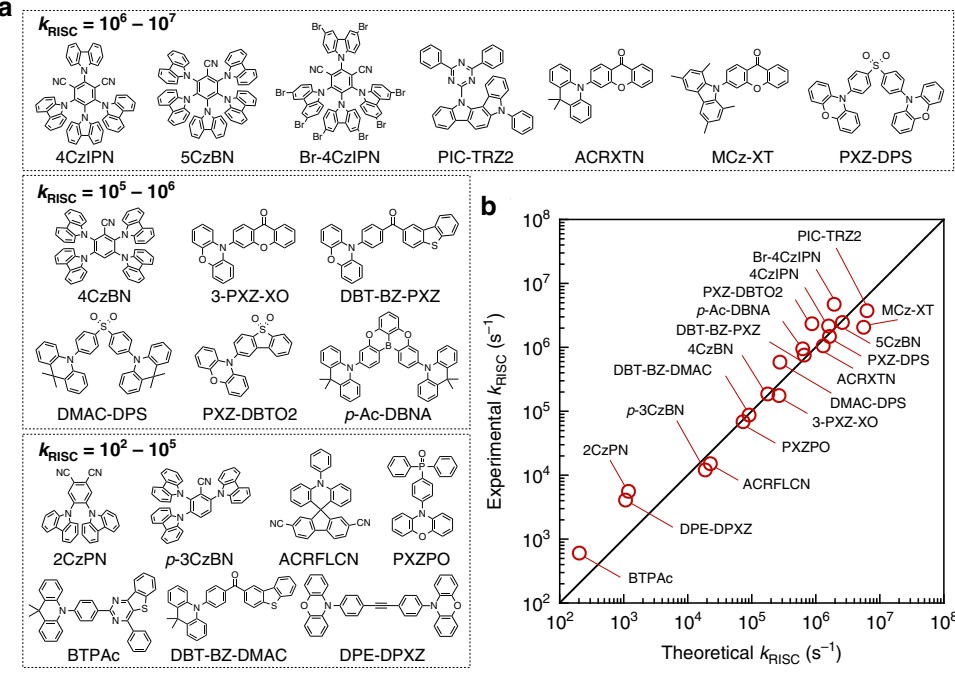

**Fig. 2 Twenty molecules examined in this study and their $k_{RISC}$ values. a** Molecular structures of the examined TADF materials categorized by their $k_{RISC}$ values. **b** Comparison of the experimental and theoretical $k_{RISC}$ values.

state ($T_2$) at the obtained MESX geometry, consistent with the model shown in Fig. 1b. This feature explains the larger errors for the parabolic approximation, which does not account for any higher-lying excited states. We attribute the uncrossed $S_1$ and $T_1$ to a nonzero exchange interaction between the singlet and triplet states, which leads to $T_1$ always lying below $S_1$ if the two states have the same electronic configuration[25]. In accordance with El-Sayed's rule[26], a large change in the orbital angular momentum between $S_1$ and $T_2$ consisting of different electronic configurations induces an effective $H_{SO}$ and thus enables spin flipping via the MESX. These results are consistent with the RISC picture anticipated based on recent theoretical and experimental studies using ACRXTN[16] and 4CzIPN[19]. It must be stressed that $S_1$–$T_2$ MESX is present in every molecule examined in this quantitative study despite their wide variety of excited-state electronic configurations, including intramolecular charge transfer (CT) states and locally excited (LE) states of $\pi$–$\pi^*$ and $n$–$\pi^*$ on either donor or acceptor units, illustrating the generality of RISC via $S_1$–$T_2$ crossing in organic donor–acceptor molecules.

**Prediction of the effects of bromination on $k_{RISC}$.** To further validate the present RISC model of Eq. (1), we computed $k_{RISC}$ of brominated analogues of representative TADF materials ACRXTN and 3-PXZ-XO: 3-(2,7-dibromo-9,9-dimethylacridan-10-yl)xanthone (Br-ACRXTN) and 3-(3,7-dibromo-phenoxazin-10-yl)xanthone (Br-3-PXZ-XO) (Fig. 3a, b). Although heavy halogen atoms such as bromine are well known to induce large $H_{SO}$ and thus facilitate ISC[27], the calculations predict that the electrophilic bromination of ACRXTN counterintuitively decreases $k_{RISC}$ from $1.3 \times 10^6$ s$^{-1}$ to $7.1 \times 10^5$ s$^{-1}$. In contrast, the bromination of 3-PXZ-XO leads to a more than hundredfold increase in $k_{RISC}$ from $2.7 \times 10^5$ s$^{-1}$ to $4.2 \times 10^7$ s$^{-1}$. Indeed, subsequent synthesis and characterization confirm the predicted opposite trend; the brominations of ACRXTN and 3-PXZ-XO caused the experimental $k_{RISC}$ to decrease from $1.0 \times 10^6$ s$^{-1}$ to $8.7 \times 10^5$ s$^{-1}$ and to increase from $1.7 \times 10^5$ s$^{-1}$ to $2.6 \times 10^7$ s$^{-1}$, respectively (see Supplementary Fig. 2 and Table S1 for details).

To the best of our knowledge, $k_{RISC}$ of over $10^7$ s$^{-1}$ for Br-3-PXZ-XO is the highest value ever reported for an organic TADF material[28]. This high $k_{RISC}$ reflects its fast transient photoluminescence decay with a delayed fluorescence lifetime of 490 ns (Fig. 3c), which is considerably shorter than typical values of several microseconds[29]. We also note that both brominated molecules exhibit similar blueshifts in their broad, unstructured CT emissions compared to the corresponding nonbrominated analogues (Fig. 3d), and this shift is attributed to the electron-withdrawing effects of the bromine atoms on the donor units, destabilizing the CT states between the donor and acceptor units (i.e., increasing the energy of the CT states).

The notable retardation of $k_{RISC}$ by bromination of ACRXTN is due to a decrease in $H_{SO}$ from 0.88 cm$^{-1}$ to 0.72 cm$^{-1}$ at the $S_1$–$T_2$ MESX geometries. This counterintuitive substitution effect of bromine on $H_{SO}$ can be rationalized by two factors. First, the $S_1$–$T_2$ spin flipping in Br-ACRXTN is compensated by a smaller change in the orbital angular momentum than that in ACRXTN (Fig. 4a, b). This is due to an increase in the occupation of the CT state in $T_2$ from 28% to 57% upon bromination, which leads to both $S_1$ and $T_2$ having similar CT states with small $H_{SO}$ according to El-Sayed's rule. Additionally, the resulting change in the orbital angular momentum of Br-ACRXTN involves the $n$ orbital of the carbonyl oxygen on the acceptor unit rather than bromine on the donor unit (Fig. 4b), suggesting that the heavy atom effect plays a minor role in determining $H_{SO}$ between $S_1$ and $T_2$. In contrast, Br-3-PXZ-XO has a perceivable contribution from the bromine atom to the orbital angular momentum change between $S_1$ of the CT state and $T_2$ of the LE $\pi$–$\pi^*$ state on the donor unit (Fig. 4c, d). Such circumstances are indeed consistent with the heavy atom effect of bromine being responsible for the increase in $H_{SO}$ from 1.2 cm$^{-1}$ to 3.5 cm$^{-1}$ and thus for the high $k_{RISC}$, over $10^7$ s$^{-1}$, in Br-3-PXZ-XO.

## Discussion

Figure 5 displays the impact of varying $H_{SO}$ and $E_A$ on $k_{RISC}$. While the existing organic TADF molecules exhibit $k_{RISC}$ smaller

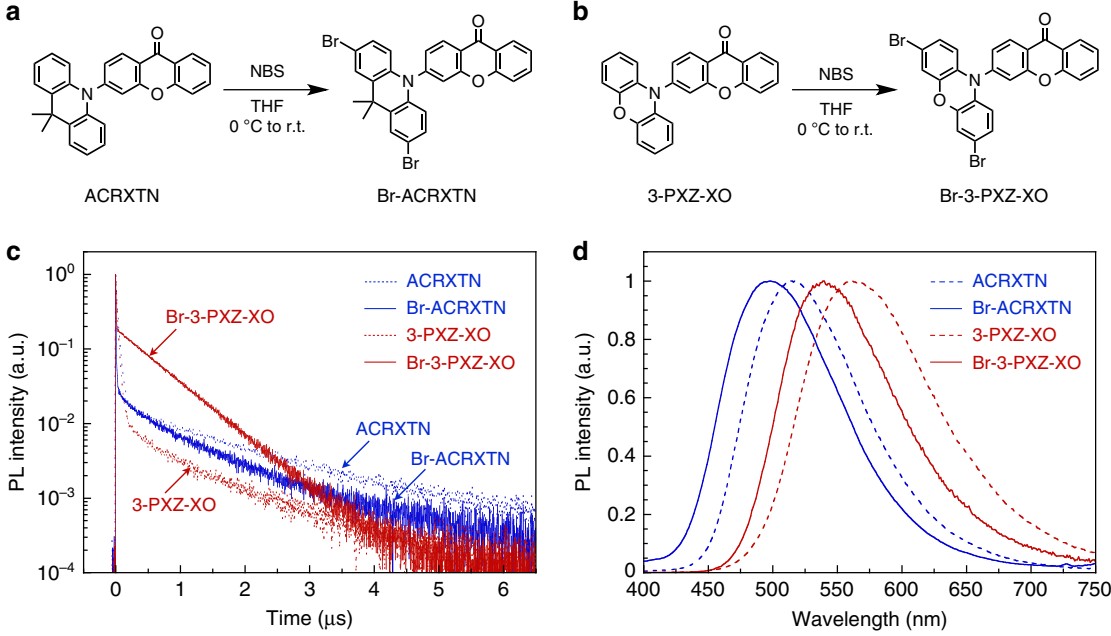

**Fig. 3 Synthesis and photoluminescence properties of the brominated molecules. a, b** Synthetic routes to Br-ACRXTN (**a**) and Br-3-PXZ-XO (**b**).
**c, d** Transient photoluminescence decays (**c**) and steady-state photoluminescence spectra (**d**) of ACRXTN, Br-ACRXTN, 3-PXZ-XO, and Br-3-PXZ-XO in a solid-state host matrix, 2,8-bis(diphenylphosphoryl)dibenzo[b,d]furan (PPF), at a doping concentration of 5 wt%.

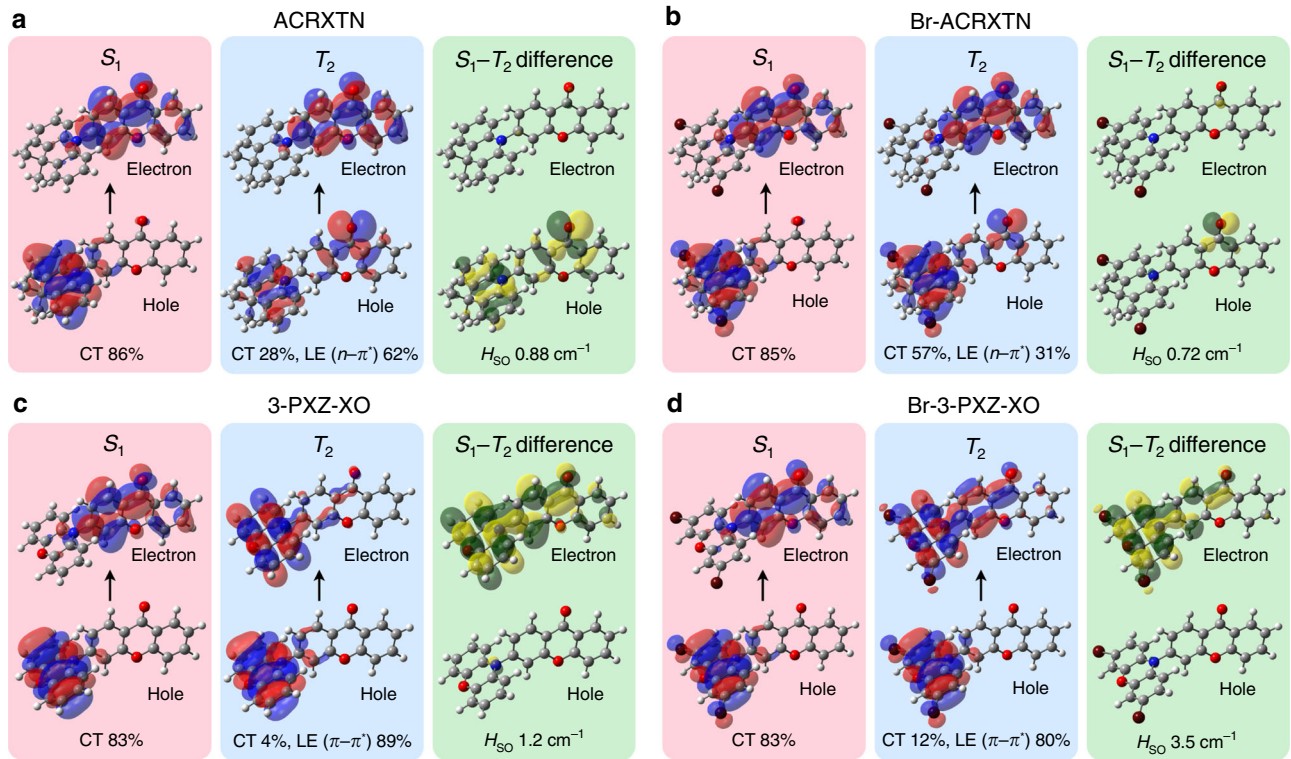

**Fig. 4 Electronic configurations of S₁–T₂ MESXs. a–d** Natural transition orbitals (NTOs) for the excited states of ACRXTN (**a**), Br-ACRXTN (**b**), 3-PXZ-XO (**c**), and Br-3-PXZ-XO (**d**) at S₁–T₂ MESX geometries. The differences in the density of the S₁ and T₂ NTOs are also shown.

than $10^8$ s$^{-1}$, the theory predicts that even $k_{RISC}$ of $10^9$ s$^{-1}$, corresponding to a time constant of 1.0 ns, can be achieved with $H_{SO}$ less than 10 cm$^{-1}$; for example, $H_{SO}$ of 7.7 cm$^{-1}$ for $E_A$ of 0.10 eV and $H_{SO}$ of 2.9 cm$^{-1}$ for $E_A$ of 0.05 eV at $T$ of 300 K. These $H_{SO}$ are an order of magnitude smaller than those of iridium-containing phosphors and could be achieved by exploiting heavy atom effects

of nonmetals in periods 3 and 4[30,31]. However, we have shown that such heuristic approaches sometimes lead to the retardation of $H_{SO}$, in part because of their more pronounced effects on the excited-state electronic configurations at the S₁–T₂ MESX geometries. Thus, for material design, a priori computational screening is essential, and the RISC model presented here allows for it.

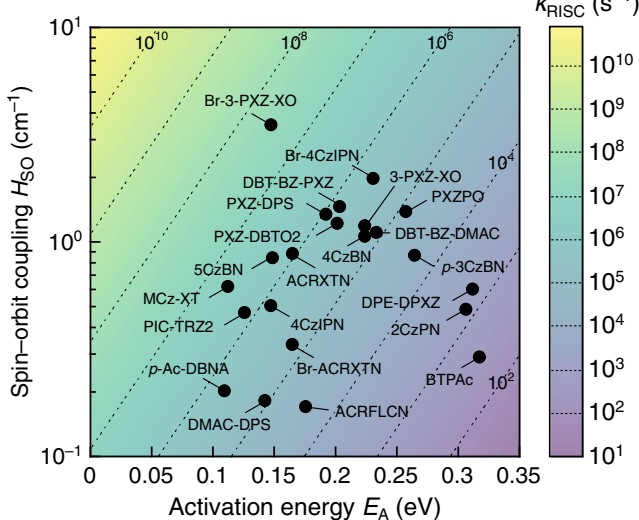

**Fig. 5 Overview of theoretical $k_{RISC}$.** $k_{RISC}$ as a function of $H_{SO}$ and $E_A$ with a fixed $\lambda$ of 0.10 eV and $T$ of 300 K. The $H_{SO}$ and $E_A$ values of the examined molecules are also plotted.

In summary, we have presented a RISC kinetic model that successfully predicts the experimental rates for a wide variety of organic TADF molecules. Our results suggest that explicitly computing the crossing seam between the singlet and triplet excited states leads to more reliable predictions than those obtained by the conventional approach using the Marcus parabolic approximation because the RISC in these molecules involves higher-lying triplet excited states. The presented model is thus a viable tool for theory-driven materials discovery with a relevant exemplar exhibiting a high $k_{RISC}$ of $2.6 \times 10^7\ \text{s}^{-1}$.

We envisage that further computational screenings of vast chemical space will facilitate the discovery of materials exploiting the spin-flipping process for various photochemical applications. We also anticipate the possible existence of materials that may not follow the model and provide platforms to discover spin-flipping mechanisms different from the presented one.

## Methods

**Computation.** The geometries of the singlet–triplet MESXs, where the square energy difference $(E_S - E_T)^2$ and the mean energy $(E_S + E_T)/2$ are minimized, were obtained by the gradient projection method[23] using a composed gradient vector **G** for the nuclear coordinates **Q**:

$$\mathbf{G(Q)} = 2(E_S(\mathbf{Q}) - E_T(\mathbf{Q}))\frac{\mathbf{v}}{|\mathbf{v}|} + \frac{1}{2}\left(\frac{\partial E_S(\mathbf{Q})}{\partial \mathbf{Q}} + \frac{\partial E_T(\mathbf{Q})}{\partial \mathbf{Q}}\right)\mathbf{P} \quad (3)$$

where

$$\mathbf{P} = 1 - \frac{\mathbf{v}\mathbf{v}^\mathrm{T}}{|\mathbf{v}|^2} \quad (4)$$

In Eq. (3), the first term contains the difference gradient vector **v** to minimize the energy difference. The second term is responsible for minimizing the mean energy, while the projection matrix **P** ensures the orthogonality between the two terms of the composed gradient vector. The excited-state energy and gradient were calculated using linear-response TDDFT with the LC-BLYP functional[32] and the 6–31+G(d) basis set within the Tamm–Dancoff approximation[24]. The range-separated parameters for the LC-BLYP functional were non-empirically optimized for each molecule to incorporate a reasonable amount of exact exchange[33,34]. The geometry optimization of the MESXs was performed with the GRRM17 program[35], which refers to the energy and gradient calculated by the Gaussian 16 program[36]. The $H_{SO}$ values were calculated perturbatively using the Breit–Pauli spin–orbit Hamiltonian with an effective charge approximation implemented in the PySOC program[37] interfaced to the Gaussian 16 program. The $E_A$ values were calculated as the electronic energy difference between the MESX and the equilibrium $T_1$. The $\lambda$ values were calculated as the difference between the $T_1$ electronic energies at the equilibrium $T_1$ and $S_1$ geometries.

**Chemical synthesis.** The synthetic procedures and characterization data of the compounds are detailed in Supplementary Information Section 1.

**Photoluminescence measurements.** Steady-state photoluminescence spectra were acquired using a Fluoromax-4 spectrophotometer (HORIBA) with 370 nm photoexcitation from a Xe arc lamp. Transient photoluminescence decay measurements were performed by time-correlated single photon counting under a flow of $N_2$ using a Fluorolog-3 fluorescence lifetime spectrometer (HORIBA) with a 370 nm LED excitation source. The absolute PL quantum yields were determined under a flow of $N_2$ using a C9920 integrating sphere system (Hamamatsu Photonics). The method for determining the experimental $k_{RISC}$ values is detailed in Supplementary Information Section 1.

## Data availability

The data that support the findings of this study available in this published Article and its Supplementary Information, or from the corresponding authors upon reasonable request.

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

## Acknowledgements

This work was supported by JST PRESTO (Grant No. JPMJPR17N1 for N.A. and JPMJPR16N8 for Y.H.), Grant-in-Aid for JSPS KAKENHI Grant (No. JP20K15252 for N.A.) and JST-ERATO (Grant No. JPMJER1903 for S.M. and Y.H.). The computations were partially performed at the computer cennter of Kyoto University and the HOKUSAI system at RIKEN.

## Author contributions

N.A. and Y.H. performed the theoretical calculations. N.A. synthesized the compounds and characterized their photoluminescence properties. S.M. and Y.-J.P. supervised the project. All authors contributed to the discussion, writing, and editing of the manuscript.

## Competing interests

The authors declare no competing interests.
