## [Peer Review File · Nature Communications]

REVIEWER COMMENTS

Reviewer #1 (Remarks to the Author):

This manuscript presents a detailed study of a library of 22 organic molecules for kinetics of reverse intersystem crossing from the triplet to the singlet excited state. Molecules were selected where the rate constants for reverse intersystem crossing span a range of 5 orders of magnitude. The study is mostly focused on computation, but also involved synthesis of some of the molecules and photophysical measurements of these synthesized molecules. The authors compared published rate constants for reversed intersystem crossing with their computation using two different models. The conventional approach using the Marcus parabolic approximation appears to be less reliable than the new approach which includes higher-lying triplet excited states. The work seems to be conducted carefully and should be novel enough to justify publication in Nature Communications. I recommend to accept this manuscript for publication after the minor corrections and suggestions listed below have been addressed.

Page 10, line 8: The authors should include what method was used for collecting the photoluminescence decays. Was it done by time-correlated single photon counting or by multi channel scaling or by another method?

Page 8, line 14: "Our results suggest that explicitly computing singlet–triplet crossing seams leads to more reliable predictions..." Something is wrong with this sentence. Delete "seams"

Table S1: The quantum yields " Φ_{PF} " and " Φ_{DF} " should be defined in the footnotes.

Reviewer #2 (Remarks to the Author):

As I'm not a theoretician, I can't judge the accuracy of the calculations and methods, nor the significance of the work in relation to the state-of-the-art in the field of computational chemistry. However, I can comment on the significance of these findings for TDA/RISC research. On the whole the paper is very interesting and shows that the prediction of reverse intersystem crossing rates should take into account higher-lying triplet excited states. Whilst the work may predict the kinetics of RISC, the paper does not lead to design rules which would have increased the impact of this work. As such, I do not believe that the paper has sufficient urgency to be published in Nature Communications, but it is a very good quality piece of work. If the advances in the theory are highlighted and commented on positively by specialist referees then these opinions should override my overall judgement on this paper.

If published, the authors should take note of the following comments:

The quality of English is not satisfactory and the paper needs thorough revision in this respect.

On page 3, I believe that the authors mean triazines instead of cyclic amines.

On page 6 the authors say that "...brominated molecules exhibit similar blue shifts in their broad, unstructured CT emissions compared to the corresponding non-brominated analogues...". However, the data show that brominated analogues are red shifted in figure 3d. When the authors say that the

bromine atoms have an electron withdrawing effect on the donor units, this will deepen the corresponding donor HOMO energy level and it should lead to the destabilisation of the CT state. Therefore, is the labelling in figure 3d correct? Also, for the broad readership of this journal, the term 'destabilising' could do with a better definition. In figure 3a, it is difficult to see which lines are which between brominated and non-brominated pairs. It would be better if the compounds show an arrow towards their respective PL decay curves.

The experimental data provided for the new compounds are fine, but the melting points of the two new solid materials are missing. These should be added to the SI experimental section.

Reviewer #3 (Remarks to the Author):

In the present manuscript NCOMMS-20-17872-T, the authors present a kinetic model to describe the reverse intersystem crossing in organic emitter molecules that show thermally activated delayed fluorescence. In this model they include an intermediate, higher-lying triplet state, i.e. T₂, that facilitates the actual spin flip. The model is tested on a variety of experimental results known from literature and further applied to newly synthesized molecules. Overall, the manuscript is very interesting for the field of organic light-emitting diode (OLED) material research and presents a methodology to include advanced computational schemes in material discovery. There are some points which I believe are important to be addressed before a recommendation can be given. My expertise as reviewer lies in the experimental section of organic electronics, so clearly I cannot judge with all the detail regarding the theoretical work. Please accept my apologies, if I ask stupid questions in this regard.

1. The quality of the English needs to be improved in multiple parts of the manuscript. I would advise the authors to undertake a careful language revision. Here, I indicate some examples:

- page 2, line 7: grammar mistake: 'Although new materials ...'

- page 3, line 17: 'for excited-state density' - I am not sure, what this is supposed to mean

2. The data from literature seems to agree very well with the simulated values of K_{RISC} . I would appreciate, if the authors can comment also in the text or SI, whether there are also materials that do not follow this agreement. If so, what are possible explanations. In addition, it would be good, if the authors can disclose their selection criteria for literature data. Are these materials randomly chosen with only requirement that all experimental data is available or do they represent some sort of subset of materials.

3. On page 2, lines 4ff: The explanation of the approach used by the authors is not made clear enough. In which model (Fig. 1a or 1b) do the authors calculate crossing seams and with this k_{RISC} ?

4. Why are the parabola in Figure 1a crossing between S₁ and T₁, while in Fig 1b, the S₁ and T₁ parabolae do not cross?

5. On page 4, line 6, the authors discuss the crossing condition as the square energy difference ($E_{\text{S}} - E_{\text{T}}$)²=0. Doesn't this imply that also the simple difference need to be zero (i.e. $E_{\text{S}} - E_{\text{T}} = 0$)?

6. Page 5, line 24: the second value stated for k_{RISC} is missing its unit (s^{-1}). Please correct.

7. Figure 3 the difference of dashed and solid lines is only hard to distinguish, in particular for the transient decays where the noise compromises the readability.

8. On page 8, the authors discuss the limit of RISC rates based on their model. Here the presentation seems inconsistent to me. In line 4 ff, the authors give the spin-orbit coupling in inverse cm, the activation energy (E_{A}) in eV. Typical values here are 7.7 cm^{-1} and 0.1 eV respectively. In their simulation (results shown in Figure 5), the authors present the orders of magnitude of k_{RISC} as a color map on a plot where the E_{A} is the x-axis and the H_{SO} the y-axis. Here the y-axis spans from 0.1 to 10 eV . If I convert the typical wavenumber given for H_{SO} in the text, this would translate to the range of meV. How does this compare in this plot (Summary: The absolute values text vs. figure for H_{SO} seem to be off substantially). Please check carefully.

9. If I am not mistaken, the newly developed materials are not all included in Figure 2b, where the correlation of experimental and simulation results is shown. These results should be included and compared. Do they fit the overall trend?

We would like to thank all the reviewers for their thoughtful and positive comments on our work. Their constructive suggestions have certainly helped us improve our manuscript. We hope that the reviewers will find the revised manuscript and the point-by-point response convincing and satisfactory to justify publication.

REVIEWER COMMENTS

Reviewer #1 (Remarks to the Author):

This manuscript presents a detailed study of a library of 22 organic molecules for kinetics of reverse intersystem crossing from the triplet to the singlet excited state. Molecules were selected where the rate constants for reverse intersystem crossing span a range of 5 orders of magnitude. The study is mostly focused on computation, but also involved synthesis of some of the molecules and photophysical measurements of these synthesized molecules. The authors compared published rate constants for reversed intersystem crossing with their computation using two different models. The conventional approach using the Marcus parabolic approximation appears to be less reliable than the new approach which includes higher-lying triplet excited states. The work seems to be conducted carefully and should be novel enough to justify publication in Nature Communications. I recommend to accept this manuscript for publication after the minor corrections and suggestions listed below have been addressed.

Re: We thank the reviewer for recognizing the novelty of our work. We also appreciate the valuable suggestions for improving our manuscript.

Page 10, line8: The authors should include what method was used for collecting the photoluminescence decays. Was it done by time-correlated single photon counting or by multi channel scaling or by another method?

Re: This was done by time-correlated single photon counting. To clarify this point, we have modified the sentence to:

“Transient photoluminescence decay measurements were performed by time-correlated single photon counting under a flow of N₂ using a Fluorolog-3 fluorescence lifetime spectrometer (HORIBA) with a 370 nm LED excitation source”.

Page 8, line 14: “Our results suggest that explicitly computing singlet–triplet crossing seams leads to more reliable predictions...” Something is wrong with this sentence. Delete “seams”

Re: We apologize for the poor choice of words and would like to somehow leave “seam” because it is one of the important concepts of this work. Thus, we modified the sentence to:

“Our results suggest that explicitly computing the crossing seam between the singlet and triplet excited states leads to more reliable predictions than those obtained by the conventional approach using the Marcus parabolic approximation”

We hope this change will address the issue of English presentation.

Table S1: The quantum yields “ Φ_{PF} ” and “ Φ_{DF} ” should be defined in the footnotes.

Re: We have modified Table S1 accordingly by adding the following footnotes for “ Φ_{PF}^d ” and “ Φ_{DF}^e ”:

^d Photoluminescence quantum yield of prompt fluorescence. ^e Photoluminescence quantum

yield of delayed fluorescence.”

Reviewer #2 (Remarks to the Author):

As I'm not a theoretician, I can't judge the accuracy of the calculations and methods, nor the significance of the work in relation to the state-of-the-art in the field of computational chemistry. However, I can comment on the significance of these findings for TDA/RISC research. On the whole the paper is very interesting and shows that the prediction of reverse intersystem crossing rates should take into account higher-lying triplet excited states. Whilst the work may predict the kinetics of RISC, the paper does not lead to design rules which would have increased the impact of this work. As such, I do not believe that the paper has sufficient urgency to be published in Nature Communications, but it is a very good quality piece of work. If the advances in the theory are highlighted and commented on positively by specialist referees then these opinions should override my overall judgement on this paper. If published, the authors should take note of the following comments:

Re: We thank the reviewer for these valuable comments/suggestions and explicitly referring to our work as “very interesting”. Based on the comments from the other reviewers, there appears to be a positive consensus on the novelty and significance of our work. Additionally, the theory and experiments showed that k_{RISC} can be increased by introducing heavy atoms on the π -conjugated unit on which T_2 is localized, and a relevant example, Br-3-PXZ-XO, exhibited increased k_{RISC} of $2.6 \times 10^7 \text{ s}^{-1}$. We believe that this result provides a concrete design rule for future materials discovery. We would appreciate it if the reviewer could take these points into account and kindly reconsider our manuscript.

The quality of English is not satisfactory and the paper needs thorough revision in this respect.

Re: The manuscript has been reviewed by a native English speaker via a language editing service. We hope the quality of the English is now satisfactory.

On page 3, I believe that the authors mean triazines instead of cyclic amines.

Re: Our intention was to refer to the triazine unit in PIC-TRZ2 and the benzo[4,5]thieno[3,2-*d*]pyrimidine unit in BTPAc as cyclic amines. To improve the clarity of this point, we have replaced “cyclic amines” with “heterocycles” on page 3, line 15.

On page 6 the authors say that “...brominated molecules exhibit similar blue shifts in their broad, unstructured CT emissions compared to the corresponding non-brominated analogues...”. However, the data show that brominated analogues are red shifted in figure 3d. When the authors say that the bromine atoms have an electron withdrawing effect on the donor units, this will deepen the corresponding donor HOMO energy level and it should lead to the destabilisation of the CT state. Therefore, is the labelling in figure 3d correct?

Re: We apologize for the mislabelling of Fig. 3d. To correct this point, we have modified Fig. 3d as shown below:

Also, for the broad readership of this journal, the term ‘destabilising’ could do with a better definition.

Re: Accordingly, we have modified the sentence to:

“destabilizing the CT states between the donor and acceptor units (i.e., increasing the energy of the CT states)”

In figure 3a, it is difficult to see which lines are which between brominated and non-brominated pairs. It would be better if the compounds show an arrow towards their respective PL decay curves.

Re: We suppose that the reviewer means Fig. 3c instead of Fig. 3a. We have added arrows to Fig. 3c, and it now appears as:

The experimental data provided for the new compounds are fine, but the melting points of the two new solid materials are missing. These should be added to the SI experimental section.

Re: We have added the melting points of the two new compounds and the experimental method in the SI.

Reviewer #3 (Remarks to the Author):

In the present manuscript NCOMMS-20-17872-T, the authors present a kinetic model to describe the reverse intersystem crossing in organic emitter molecules that show thermally activated delayed fluorescence. In this model they include an intermediate, higher-lying triplet state, i.e. T₂, that facilitates the actual spin flip. The model is tested on a variety of experimental results known from literature and further applied to newly synthesized molecules. Overall, the manuscript is very interesting for the field of organic light-emitting diode (OLED) material research and presents a methodology to include advanced computational schemes in material discovery. There are some points which I believe are important to be addressed before a recommendation can be given. My expertise as reviewer lies in the experimental section of organic electronics, so clearly I cannot judge with all the detail regarding the theoretical work. Please accept my apologies, if I ask stupid questions in this regard.

Re: We were pleased by the positive response from the reviewer. We also appreciate these valuable suggestions, which have certainly improved our manuscript.

1. The quality of the English needs to be improved in multiple parts of the manuscript. I would advise the authors to undertake a careful language revision.

Re: The manuscript has been reviewed by a native English speaker via a language editing service. We hope the quality of the English is now satisfactory.

Here, I indicate some examples:

- page 2, line 7: grammar mistake: 'Although new materials ...'

Re: We have corrected the grammatical mistake by adding "been" to the sentence. The modified sentence now reads:

"Although new materials have typically been discovered experimentally, a fundamental understanding of RISC kinetics and strategy for predicting the rate constants may open vast opportunities for theory-driven materials discovery."

- page 3, line 17: 'for excited-state density' - I am not sure, what this is supposed to mean

Re: Our intention was that "excited-state density" means the population density (or concentration) of excitons. To clarify this point, we have modified the sentence to:

"We collected literature values for the steady-state and transient photoluminescence data of these molecules and estimated their k_{RISC} values from differential rate equations of the population densities of their singlet and triplet excited states"

2. The data from literature seems to agree very well with the simulated values of K_{RISC} . I would appreciate, if the authors can comment also in the text or SI, whether there are also materials that do not follow this agreement. If so, what are possible explanations.

Re: This is a very interesting point. We have not found such materials yet, but are now trying to find them because they may show a unique RISC mechanism that is different from the proposed one and may possibly surpass the conventional materials. To state this point, we have added the following sentence as the last sentence in the summary of the text:

“We also anticipate the possible existence of materials that may unfollow the model and provide platforms to discover novel spin-flipping mechanisms.”

In addition, it would be good, if the authors can disclose their selection criteria for literature data. Are these materials randomly chosen with only requirement that all experimental data is available or do they represent some sort of subset of materials.

Re: We intended to make the list diverse in terms of k_{RISC} . The number of donor–acceptor TADF materials that have been reported to exhibit low $k_{\text{RISC}} < 10^5 \text{ s}^{-1}$ is limited, and the compounds were thus chosen with no other specific intent. For high $k_{\text{RISC}} > 10^5 \text{ s}^{-1}$, we chose representative TADF materials, such as 4CzIPN and ACRXTN, that have been studied by many groups. Additionally, we selected compounds such that they had various popular acceptor units for TADF materials, such as heterocycles, aryl nitriles, ketones, boranes, sulfones, and phosphine oxides. We have included this information on page 7 in the SI.

3. On page 2, lines 4ff: The explanation of the approach used by the authors is not made clear enough. In which model (Fig. 1a or 1b) do the authors calculate crossing seams and with this k_{RISC} ?

Re: We suppose that the reviewer means page 5 instead of page 2 since page 2 does not include an explanation of our approach to calculations. We have included the information that we used the conventional model as depicted in Fig 1a. The modified sentence on page 5, lines 1–3 now reads:

“a larger MALE of 1.2, corresponding to an error of 1.2 orders of magnitude, is observed for the values based on the conventional model shown in Fig. 1a and the parabolic approximation of Equation (2)”

For clarity, we have also modified a sentence on page 5, line 5, by specifying that Fig 1b is consistent with the calculation results suggesting that S_1 does not cross T_1 and instead crosses T_2 . The modified sentence now reads:

“Closer inspection of the data further reveals that the lowest singlet excited state (S_1) does not cross the lowest triplet state (T_1) and instead crosses the higher-lying triplet state (T_2) at the obtained MESX geometry, consistent with the model shown in Fig. 1b”

4. Why are the parabola in Figure 1a crossing between S_1 and T_1 , while in Fig 1b, the S_1 and T_1 parabolae do not cross?

Re: As shown in Fig. 1a, which represents the conventional model of direct RISC between S_1 and T_1 , the parabolas of S_1 and T_1 cross. On the other hand, as shown in Fig. 1b, for the proposed RISC model, S_1 does not cross T_1 and instead crosses T_2 . On page 5, line 8, we attributed the uncrossed S_1 and T_1 to the nonzero exchange interaction between the singlet and triplet states, which leads to T_1 always lying below S_1 if the two states have the same electronic configuration.

5. On page 4, line 6, the authors discuss the crossing condition as the square energy difference $(E_S - E_T)^2 = 0$. Doesn't this imply that also the simple difference need to be zero (i.e. $E_S - E_T = 0$)?

Re: Yes, the simple energy difference is 0 at the crossing seam. The square energy difference, $(E_S - E_T)^2$, is a convex parabola and thus convenient for finding the minimum of $|E_S - E_T|$ by gradient calculations. To avoid confusion, we have modified the sentence to

“we employed a constrained optimization algorithm using the gradient projection method²³, which minimizes the mean energy of the singlet and triplet states $(E_S + E_T)/2$ while simultaneously fulfilling the crossing condition of the energy difference $E_S - E_T = 0$ ”

6. Page 5, line 24: the second value stated for k_{RISC} is missing its unit (s^{-1}). Please correct.

Re: We have modified the sentence by adding the unit. The modified sentence now reads:

“the calculations predict that the electrophilic bromination of ACRXTN counterintuitively decreases the k_{RISC} from $1.3 \times 10^6 \text{ s}^{-1}$ to $7.1 \times 10^5 \text{ s}^{-1}$ ”

7. Figure 3 the difference of dashed and solid lines is only hard to distinguish, in particular for the transient decays where the noise compromises the readability.

Re: This issue was also pointed out by reviewer #2. According to the suggestion from reviewer #2, we have modified Fig. 3 to improve the readability. Fig. 3d was modified to the following:

8. On page 8, the authors discuss the limit of RISC rates based on their model. Here the presentation seems inconsistent to me. In line 4 ff, the authors give the spin-orbit coupling in inverse cm, the activation energy (E_A) in eV. Typical values here are 7.7 cm^{-1} and 0.1 eV respectively. In their simulation (results shown in Figure 5), the authors present the orders of magnitude of k_{RISC} as a color map on a plot where the E_A is the x-axis and the H_{SO} the y-axis. Here the y-axis spans from 0.1 to 10 eV . If I convert the typical wavenumber given for H_{SO} in the text, this would translate to the range of meV. How does this compare in this plot (Summary: The absolute values text vs. figure for H_{SO} seem to be off substantially). Please check carefully.

Re: We apologize for the mislabelling of the unit of the spin-orbit coupling in Fig 5. The correct unit is cm^{-1} . Fig. 5 has been modified to the following:

9. If I am not mistaken, the newly developed materials are not all included in Figure 2b, where the correlation of experimental and simulation results is shown. These results should be included and compared. Do they fit the overall trend?

Re: Yes, they fit the overall trend as shown in the figure below. We have added this figure as a new figure, Fig. S2 in the SI.

REVIEWERS' COMMENTS:

Reviewer #1 (Remarks to the Author):

The authors addressed my comments in the revised manuscript satisfactorily and I therefore recommend to accept the revised manuscript for publication.

However, the authors should correct one phrase, which was added in the revision:

Page 10, line 1: "... The possible existence of materials that may unfollow the model..."

The word "unfollow", which is usually used in context of social media, should not be used here.

Using "not follow" would be better.

Reviewer #2 (Remarks to the Author):

I am satisfied with the responses to the referees' comments and in my opinion this version of the manuscript can be published.

Reviewer #3 (Remarks to the Author):

In the present, NCOMMS-20-17872A, the authors have provided a detailed revision of the original manuscript based on three individual review reports. All points that I have stated in the first round of review have been addressed in a satisfactory fashion. I am happy to recommend acceptance of this manuscript in Nature Communication.

REVIEWERS' COMMENTS:

Reviewer #1 (Remarks to the Author):

The authors addressed my comments in the revised manuscript satisfactorily and I therefore recommend to accept the revised manuscript for publication. However, the authors should correct one phrase, which was added in the revision: Page 10, line 1: "... The possible existence of materials that may unfollow the model..." The word "unfollow", which is usually used in context of social media, should not be used here. Using "not follow" would be better.

Re: We have modified this sentence accordingly. The modified sentence now reads: "We also anticipate the possible existence of materials that may not follow the model and provide platforms to discover novel spin-flipping mechanisms."